# A Novel PLLA/MgF_2_ Coating on Mg Alloy by Ultrasonic Atomization Spraying for Controlling Degradation and Improving Biocompatibility

**DOI:** 10.3390/ma16020682

**Published:** 2023-01-10

**Authors:** Wenpeng Peng, Yizhe Chen, Hongde Fan, Shanshan Chen, Hui Wang, Xiang Song

**Affiliations:** 1Cardiovascular Center, The Fourth Affiliated Hospital of Harbin Medical University, Harbin 150000, China; 2Department of Clinical Medicine, Harbin Medical University, Harbin 150000, China; 3Union Hospital, Tongji Medical College, Huazhong University of Science and Technology, Wuhan 430022, China; 4Hubei Key Laboratory of Advanced Technology for Automotive Components, Wuhan University of Technology, Wuhan 430070, China; 5Institute of Metal Research, Chinese Academy of Sciences, Shenyang 110016, China

**Keywords:** biodegradable, magnesium alloys, poly-l-lactic acid, hydrofluoric acid, degradation, biocompatibility

## Abstract

Problems of rapid degradation and poor biocompatibility (endothelialization and hemocompatibility) limit magnesium (Mg) alloy’s further applications in vascular stents. To solve these problems, a novel composite coating was designed on Mg alloy via a two-step method. First, a Mg alloy sample was immersed in hydrofluoric acid. Then, a poly-l-lactic acid (PLLA) coating was made by ultrasonic atomization spraying with 5 and 10 layers (referred to as PLLA(5)-HF-Mg and PLLA(10)-HF-Mg). Characterizations were analyzed from the microstructure, element distribution, and wettability. The degradation behavior was tested with an electrochemical test and immersion test. Endothelialization was investigated using human umbilical vein endothelial cells (HUVECs). Hemocompatibility was examined with a platelet adhesion test. The results showed that the PLLA coating could not only cover the surface, but also could permeate through and cover the holes on the MgF_2_ layer, mechanically locked with the substrate. Thus, the composite coating had higher corrosion resistance. The PLLA/MgF_2_ coating, especially on PLLA(10)-HF-Mg, enhanced HUVECs’ viability and growth. While incubated with platelets, the PLLA/MgF_2_ coating, especially on PLLA(10)-HF-Mg, had the lowest platelet adhesion number and activity. Taken together, the novel PLLA/MgF_2_ coating controls Mg alloy’s degradation by spraying different layers of PLLA, resulting in better endothelialization and hemocompatibility, providing a promising candidate for cardiovascular stents.

## 1. Introduction

Coronary heart disease (CHD) is still the leading cause of death in the world with a high incidence and mortality rate [1]. Coronary stent deployment is a critical implement in the management of CHD. Coronary stents have undergone constant reform from the first generation of percutaneous transluminal coronary angioplasty (PTCA) to drug-eluting stents (DESs) and bioabsorbable stents (BRSs) [2,3]. Now, DESs are the first choice for the treatment of CHD in interventional cardiology practice. The stent materials used are mainly stainless steel, titanium alloy, cobalt-based alloy, and other traditional metal materials. However, these materials cannot be completely degraded, and their long-term existence in the body after implantation will cause many complications, such as inflammation of tissues around the stent, the problem of re-endothelialization, and late persistent or acquired struts’ malposition, resulting in late stent restenosis [4]. It is hard to insert another stent after inserting a permanent metal stent. The answer to these issues leads to the fourth generation of BRSs, which remains in development and attracts researchers. BRS materials should not only maintain the physical and biological properties of DESs, but also be able to be fully absorbed, leaving the vessel with a healthy endothelium and normal vasomotion [5]. At present, BRSs mainly include metal degradable stents and polymer degradable stents. Polymers are used more often in biodegradable scaffolds. The typical polymer degradation stents are Absorb BVS [6] and Neovas [7]. However, there are still some shortcomings, such as inadequate support and slow degradation and other problems.

Magnesium (Mg) alloy is one of the most potential metal degradable materials of stents [8]. Mg has both good mechanical properties and good biological properties. Mg alloy can completely degrade, and the degradation products can be discharged through metabolism. It has no obvious toxicity to the surrounding tissues [9]. Furthermore, the degradation product of Mg alloy is the cofactor of adenosine triphosphate (ATP), as well as a physiological calcium antagonist, which can effectively prevent the injury of calcium overload caused by various ischemia–reperfusion injuries. While the concentration of Mg^2+^ in the human body is 0.7~1.0 mmol/L [10], the amount of Mg^2+^ released by a Mg alloy stent is very small. Although Mg has excellent biodegradable prospects, Mg alloy shows a relatively high rate of degradation, which cannot provide reliable mechanical support for the vessel [11]. For example, a Mg-based stent, AMS-1(Biotronik, Berlin, Germany), degraded completely after being implanted for 4 months [12]. DREAMS-2G, which had been marketed, was clinically detected to be absorbed completely at 12 months by intravenous ultrasound (IVUS) and optical coherence tomography (OCT) [13]. Degrading too quickly produces too much hydrogen, which has cytotoxicity [14]. Rapid degradation causes an increase in the surface area, which in turn could lead to increased plasma protein adsorption, and subsequent platelet adhesion and activation. These may trigger the coagulation cascade and initiate thrombosis [15].

To overcome these limitations, recent advances in surface modification and coatings significantly reduce the degradation rate [5]. Studies show that surface modifications are very effective methods, including sol–gel, micro-arc oxidation [16,17], chemical conversion, polymer coatings, and so on [18,19]. These surface modifications can control the Mg degradation rate and biocompatibility without altering the bulk properties of Mg alloy. Furthermore, modified surfaces or coatings degrade gradually in vivo without any adverse effect on the surrounding tissues [18]. Among them, fluorine conversion treating on Mg alloy has proved to be an effective method for corrosion resistance with the occurrence of MgF_2_. MgF_2_ has been proved to slow down the degradation rates of Mg alloys [20,21], thus preventing hydrogen accumulation in tissues and fluid [22]. Moreover, numerous studies showed MgF_2_ had no cytotoxicity [23]. However, it has proved difficult for endothelial cells to colonize [24]. Thus, the poor endothelialization of MgF_2_ has limited its application.

A polymer coating on Mg alloy can solve the problems of rapid degradation and poor endothelialization, such as poly-l-lactic acid (PLLA), poly lactic-co-glycolic acid (PLGA), and poly caprolactone (PCL) [25]. PLLA can be metabolized via the Krebs cycle into carbon dioxide and water without toxic degradation products [26]. It has proved highly biodegradable and biocompatible [27]. It is illustrated to be safe in humans [28]. Now, it is a clinically approved material. It is widely used in biomedical applications such as bioabsorbable polymer coatings [29], sutures, orthopedic fixation screws, and tissue engineering scaffolds [30,31]. However, a famous bioabsorbable stent, Absorb BVS, showed the shortcomings of PLLA, such as hemodynamic changes caused by thick thickness, inadequate mechanical properties, and too-slow degradation rate [32]. Considering the above, it is a good choice to combine the composite PLLA/MgF_2_ coating with Mg alloy. In this composite coating, MgF_2_ can serve as an appropriate coupling agent between the PLLA coating and Mg alloy to effectively strengthen the adhesion.

In the process of fabricating a PLLA coating, there are various methods, such as dip coating, spin coating, ultrasonic atomization spraying, and so on [33,34]. Among them, the ultrasonic atomization spraying method can make the surface coating thinner and more stable. Atomized particles of PLLA can reach 1–3 μm, and this makes it easy to control the thickness, since the thickness of PLLA coatings affects the degradation time. Meanwhile, this method is of high material utilization, reducing the waste of materials [35,36]. It can be predicted that the combination of PLLA and MgF_2_ by ultrasonic atomization spraying will be effective in controlling degradation and improving the biocompatibility of Mg alloy. However, reports are scarce. Further study on the effect of this novel composite coatings on Mg alloy is important for promoting the development of metal degradable stents.

In this study, taking advantage of a MgF_2_ and PLLA coating, a novel composite coating with low degradation and good biocompatibility was made on Mg alloy. The Mg alloy was treated with hydrofluoric acid and then fabricated with PLLA coatings by ultrasonic atomization spraying. To better illustrate the advantage of the PLLA coating by ultrasonic atomization spraying, the PLLA coating was sprayed with 5 and 10 layers. The characterization was analyzed from the microstructure, element composition, and surface wettability. The biodegradation property was investigated using electrochemistry and a long-term immersion test in vitro. Endothelialization was systematically investigated using human umbilical vein endothelial cells (HUVECs). Hemocompatibility was examined with a platelet adhesion test.

## 2. Materials and Methods

### 2.1. Preparation of Mg Substrate

As Figure 1 shows, an AZ31 alloy sheet (as-rolled, 2 mm thick, Dongguan Jubao Magnesium alloy Material Co., Ltd., Dongguan, China) was cut into 10 mm × 10 mm squares using electrical discharge machining and used as the biodegradable model substrate. Mg squares were degreased in acetone for 15 min using an ultrasonic cleaner, and then cleaned in 100% ethanol for 15 min by an ultrasonic cleaner. These can remove grease and impurities from the surface. Mg squares were then dried in air. All Mg substrates were polished with 600, 1200, and 2000 grit SiC paper to ensure consistency and then cleaned again using 100% ethanol for 15 min in the ultrasonic cleaner, then dried in a vacuum-drying oven (referred to as bare Mg). This can remove impurities from the polishing process. Thus, the surface of all samples was smooth and clean, which was conducive to the production of the fluorine conversion coating and the spraying of PLLA.

### 2.2. Fabrication of MgF_2_ and Polymer Coatings

The preparing process of samples is shown in Figure 1. Polished Mg substrates were soaked in hydrofluoric acid of concentration 40% at room temperature for 48 h and then dried in a vacuum-drying oven for 24 h (referred to as HF-Mg). Some HF-Mg samples were coated with a polymer. The polymer selected for this study was PLLA (Nature Works 4032D). PLLA with a mass of 1 g was dissolved in 64 g methylene chloride to achieve the concentration of wt.% 1.5%. The processing of dissolving PLLA was accelerated using a speed mixer at 250 rpm. Then, the PLLA solution was sprayed on the Mg substrate using an ultrasonic atomization spraying device (Hangzhou Jiazhen Ultrasonic Technology Co., LTD, HC-LAKSH-GL, Hangzhou, China) with 5 and 10 layers (referred to as PLLA (5)-HF-Mg and PLLA (10)-HF-Mg). The coated substrates were dried in a vacuum-drying oven for 24 h.

### 2.3. Surface Characterization

Scanning electron microscopy (SEM, Hitachi Regulus8100, Tokyo, Japan) was used to analyze the influence of different coatings on surface morphology. To better reveal the coating, especially the thickness of the coating, the cross-sections of samples were observed by SEM. The element types and contents were analyzed by energy-dispersive X-ray spectroscopy (EDS, Hitachi Regulus8100).

The water contact angle of samples can be measured using a contact angle goniometer (Kino SL200B) at room temperature, which reflects the surface wettability. All samples were fixed on the slide; 1 drop of water was dropped on the surface, and the image of the droplet on the sample was captured. Then, the water contact angle was analyzed in the drop shape analysis software named Kruss.

### 2.4. In Vitro Biodegradation

In vitro biodegradation behaviors were examined in a three-electrolyte cell (working electrode, a platinum counter electrode, and a saturated calomel reference electrode) using the CHI660E workstation in simulated body fluid (SBF) [37] solution at 37 °C. The exposed area of the working electrode to the electrolyte was 1 cm^2^. A stable open circuit potential (OCP) was established for 10 min prior to electrochemical impedance spectroscopy (EIS) testing. Potentiodynamic polarization (PDP) curves were obtained by scanning from −3 V to 0 V at a scan rate of 5 mV/s. The corrosion potential (E_corr_) and corrosion current density (I_corr_) were determined by extrapolating the tangent line of the anodic half and cathodic half, following the ASTM G102-89 standard.

Immersion testing can also be examined by long-term corrosion resistance and in vitro biodegradation. The samples were immersed in SBF for 15 days. The sBF solution is the most commonly used solution for in vitro simulation. Its pH value, ionic composition, and concentration are close to that of human body fluids. In this study, the method published by Tadashi [38] was used to prepare the SBF solution. Table 1 shows the reagents needed to prepare 1 L of SBF solution, adding sequence and quantity. The pH value of the immersion medium was monitored with a PH meter every day. Then, the morphology of the samples’ surface and element composition were examined using SEM and EDS.

### 2.5. Cell Culture and Tests

HUVECs (cell line, Wuhan Punosei Life Technology, Wuhan, China) were cultured in DMEM (high glucose) with 10% FBS, 100 U/mL penicillin, and 100 mg/mL streptomycin. The cell-testing process of the coated and bare Mg alloy is shown in Figure 2. As Figure 2A shows, each kind of sample was settled in the bottom of cell culture dishes. Then, 1 mL cell suspensions were seeded on surfaces of samples at a density of 2 × 10^4^ cells/cm^2^. The samples were, respectively, taken out after 24 h of incubation and rinsed with PBS. The samples were then immersed in calcein AM/PI (Beyotime Biotechnology, Shanghai, China) for 30 min and examined using a fluorescence microscope (Eclipse Ti, Nikon, Tokyo, Japan). Cell viability was measured using the cell counting kit-8 (CCK-8, Beyotime Biotechnology, Shanghai, China), as Figure 2B shows. The extract liquid was prepared by immersing samples in DMEM medium. The volume of the solution was calculated based on the volume/area of 1 mL/1.25 cm^2^, according to ISO 10993-12. The immersing time was 24 h and 72 h. HUVECs were initiated at a density of 1 × 10^4^ cells per 96 wells and incubated in a 5% CO_2_ incubator at 37 °C for 24 h. After attachment, 100 μL of extract liquids was added to each well and incubated in a 5% CO_2_ incubator at 37 °C for 24 h. After that, the culture liquid was taken out, and 100 μL medium containing 10% CCK-8 was added to each well. After an incubation of 3 h (37 °C, humidified 5% CO_2_ atmosphere), the optical density (OD) at 450 nm was determined by an ELIS multiplate reader (Sunrise, Tecan, Männedorf, Switzerland). The background was adjusted using 100 μL of each medium, containing 10% CCK-8 without cells.

### 2.6. Platelet Adhesion Test

Fresh rabbit blood was centrifuged at 3600 rpm for 20 min. The upper platelet-rich plasma was collected. Plasma was dropped onto the surface of all samples and placed in a cell culture plate. After incubation for 1 h at 37 °C, they were carefully rinsed with PBS (pH 7.2) and immersed in 2.5 wt% glutaraldehyde for 1 h at room temperature. Then, all samples were dehydrated with 50%, 60%, 70%, 80%, 90%, and 100% gradient ethanol for 10 min each. Finally, they were dried in a vacuum low-temperature dryer and characterized by SEM.

## 3. Results

### 3.1. Surface Morphology of Coated and Bare Mg Alloy Samples

The surface morphology observed using SEM is shown in Figure 3; great changes were found on the surface of the Mg alloy after polishing, fluorine transformation treating, and the ultrasonic atomization spraying of PLLA. The surface of HF-Mg was very smooth and compact. However, small holes on the fluorine conversion coating could be seen, as shown in the high-power field. Figure 3c,d show the surface morphology of PLLA (5)-HF-Mg and PLLA (10)-HF-Mg. It can be seen that the PLLA coating of PLLA (5)-HF-Mg could not cover the Mg substrate, but the PLLA coating of PLLA (10)-HF-Mg could almost cover the surface without excessive droplet overlap.

The elemental composition of the surfaces obtained by the EDS test is shown in Figure 4. Figure 4a shows the elemental composition of the bare Mg alloy. The mass fraction of Mg is 89.06%. The mass fraction of O is 3.01%, which is due to the oxidation of Mg. Figure 4b shows the elemental composition of the HF-Mg surface. The main elements were Mg and F, with a mass fraction of 32.58% and 45.5%. This indicates MgF_2_ was successfully prepared. Figure 4c,d show the elemental composition of the surfaces of PLLA (5)-HF-Mg and PLLA (10)-HF-Mg. It can be seen that the mass fraction of C and O increased greatly, which indicates the PLLA coating was successfully fabricated. Comparing Figure 4c,d, it can be seen that there was less Mg and F and more C on the surface of the PLLA (10)-HF-Mg sample. This indicates the PLLA coating with 10 layers better covered the sample surface, consistent with the result of SEM.

Figure 5 shows the thickness of the cross-section of three coated samples analyzed using SEM. Figure 5a shows the micrograph of the cross-section of HF-Mg. The thickness of the MgF_2_ layer was 0.9 μm. Figure 5b shows the micrograph of the cross-section of PLLA (5)-HF-Mg. The thickness of the PLLA with five layers was 1.1 μm. Figure 5c shows the micrograph of the cross-section of PLLA (10)-HF-Mg. The thickness of the PLLA with 10 layers was 1.5 μm, not twice as much as PLLA (5)-HF-Mg.

Surface wettability can be reflected by the water contact angle. As Figure 6 shows, the average contact angle of the bare Mg alloy surface was 73.5°, and that of the HF-Mg surface was 33.95°. Compared with the bare Mg alloy, the diffusion of water droplets was more extensive on the surface of HF-Mg. The average contact angle of PLLA (5)-HF-Mg and PLLA (10)-HF-Mg was 91.93° and 103.18°, which significantly increased compared with the bare Mg alloy and HF-Mg. It can be seen that the water droplets on the surface of PLLA (5)-HF-Mg and PLLA (10)-HF-Mg were round. Therefore, their surface is more hydrophobic than the bare Mg alloy and HF-Mg.

### 3.2. In Vitro Degradation

#### 3.2.1. Electrochemical Test

The corrosion behavior of the coated samples and bare Mg alloy was evaluated with an electrochemical test in SBF solution. The corrosion properties are demonstrated by typical polarization curves in Figure 7 and the corrosion current density (I_corr_) is in Table 2. Corrosion potential measures the thermodynamic parameters of a material in the process of electrochemical corrosion. It represents the tendency of the material to undergo electrochemical corrosion. Meanwhile, corrosion current measures the kinetic parameters and refers to the electrochemical corrosion rate. When the corrosion potential is higher and the corrosion current density is lower, the corrosion resistance of the material is better. Figure 7 shows that the PDP curves of HF-Mg, PLLA (5)-HF-Mg, and PLLA (10)-HF-Mg shifted toward lower current densities compared with bare Mg. As shown in Table 2, the current density of all three coated Mg alloys was significantly lower than that of the bare Mg alloy. The current density of PLLA (5)-HF-Mg and PLLA (10)-HF-Mg was comparatively lower than that of HF-Mg. However, there was not much difference between PLLA (10)-HF-Mg and PLLA (5)-HF-Mg.

#### 3.2.2. Immersion Test

The pH value of the medium for the coated and bare Mg alloy after immersion in SBF for 15 days is shown in Figure 8. It can be seen that the pH values of the medium for bare Mg alloy were in the range of 8.0–9.0, which were higher than that of HF-Mg, PLLA (5)-HF-Mg, and PLLA (10)-HF-Mg. The pH values of the medium for HF-Mg were in the range of 7.6–8.4, which also changed largely and were higher than that of PLLA (5)-HF-Mg and PLLA (10)-HF-Mg. However, the pH values of the medium for PLLA (5)-HF-Mg and PLLA (10)-HF-Mg were in the range of 7.6–7.8, and the change was very small.

The surface morphology after immersion in SBF for 15 days observed using SEM is shown in Figure 9. Figure 9a shows the corrosion morphology of the bare Mg alloy. The surface was uneven. There were numerous and deep cracks on the surface. Figure 9b shows the corrosion morphology of HF-Mg, with little corrosion cracks on the substrate. Figure 9c shows the corrosion morphology of PLLA (5)-HF-Mg, with a broken PLLA layer on the surface. From Figure 9d, it can be seen that there was still a PLLA layer on the surface of PLLA (10)-HF-Mg and nearly no substrate can be seen.

The element content of the corrosion surface of all samples after immersion in SBF for 15 days analyzed by EDS is shown in Figure 10. In this study, the bare Mg alloy’s surface had the highest content of O of 52.96%. On the corrosion surface of HF-Mg, the content of F was 27.93%, which was the highest. The content of F was low on the surface of PLLA (5)-HF-Mg and was almost zero on the surface of PLLA (10)-HF-Mg. While the main element in PLLA is C and O, the content of C is high on the surface of PLLA (5)-HF-Mg and PLLA (10)-HF-Mg.

### 3.3. Cell Culture and Test

#### 3.3.1. Cell Adhesion and Proliferation

The adhesion and proliferation of HUVECs after 24 h culture on the surface of Mg alloy observed by a fluorescence microscope are shown in Figure 11. Figure 11a shows the fluorescence image of HUVECs attached on the bare Mg alloy, with nearly no HUVECs and large number of dead cells. Figure 11b shows the fluorescence image of HUVECs attached on HF-Mg. There were fewer cells on the surface of HF-Mg than the bare Mg alloy. Meanwhile, from Figure 11c,d, it was found that HUVECs grew better on the PLLA/MgF_2_ coating. In particular, PLLA (10)-HF-Mg had the best performance. It had the highest HUVEC density and nearly no dead cells, which indicates that the PLLA/MgF_2_ coating is more beneficial to endothelial cell colonization.

#### 3.3.2. Cell Vitality

The relative growth rate (RGR, which represents cell vitality) of HUVECs in different extracts is shown in Figure 12. It can be seen that vitalities of HUVECs cultured in the coated-alloy extract for 1 day and 3 days were almost improved compared with the bare Mg alloy. After extracting for 1 day, the RGR of HUVECs in the bare Mg alloy extract was 24.5%. It was not increased much in the HF-Mg extract, but significantly increased in PLLA (10)-HF-Mg extracts. In the PLLA (10)-HF-Mg extract, the RGR of HUVECs was 42.5%, increasing by 73% compared with that of the bare Mg alloy. After 3 days of extracting, there was not much difference in the RGR of HUVECs in the PLLA (5)-HF-Mg and HF-Mg extracts compared with that of the bare Mg alloy. The RGR of HUVECs in the PLLA (10)-HF-Mg extract was significantly increased by 111% compared with that of the bare Mg alloy.

### 3.4. Platelet Adhesion Test

The morphology of platelets assembled onto the surface of coated and bare Mg alloy samples is shown in Figure 13A. The number of adhered platelets is shown in Figure 13B. The degree of platelet activation was assessed in terms of platelet morphology. The less the distortion and the adhesion of platelets to the material surface, the less the probability of blood coagulation and the better the blood compatibility [39]. Platelets were spread, abundant, and collected together on the surface of the bare Mg alloy, as Figure 13(Aa) shows, while there was a small number of platelets adhered to the surface of HF-Mg (Figure 13(Ab)) and PLLA (5)-HF-Mg (Figure 13(Ac)). Additionally, these platelets were dendritic and round. On the surface of PLLA (10)-HF-Mg, there was a rare platelet, and the shape was nearly round (Figure 13(Ad)).

## 4. Discussions

### 4.1. The Novel Composite Coating Controls the Degradation Rate of Mg Alloy

When a Mg substrate is immersed in hydrofluoric acid, many corrosive microcells can form on the surface of the sample.

Anodic reaction:Mg → Mg^2+^ + 2*e*^−^


Cathodic reaction:2H^+^ + 2*e*^−^ → H_2_ ↑

Additionally, other reactions:Mg^2+^ + 2F^−^ → MgF_2_ ↓
Mg^2+^ + 2OH^−^ → Mg(OH)_2_ ↓
Mg(OH)_2_ → MgO ↓ + H_2_O

Therefore, the main ingredient of fluoride conversion is MgF_2_, and some MgO, which is also shown in the result of EDS in Figure 4. From the SEM image of HF-Mg, it was found that the MgF_2_ layer was smooth and compact as shown in Figure 3 and Figure 5. Lisitsyn et al. found that the functional properties and degradation of MgF_2_ are highly dependent on point structural defects. Even in crystals, point defects play a significant role, and in many cases are simply decisive [40,41]. The holes in Figure 3b may be caused by hydrogen, which is consistent with a previous study [42]. The PLLA coating can permeate through and cover these holes, mechanically locked with the substrate. This provides a good physical barrier and reduces the corrosion of the Mg alloy substrate. In this experiment, ultrasonic atomization spraying proved to be a useful method for polymer coating, consistent with previous studies. For example, in Chen and Pan’s study [43], a PLGA coating was fabricated on 316 L stainless-steel stents using the ultrasonic atomization spraying method. The coating was found to be not only smooth and uniform, but also could withstand compressive strain and tensile strain. Thus, the stent would not crack in the expansion process [44]. In this study, it can be seen in Figure 3 that the PLLA droplets could not completely cover the Mg alloy substrate of PLLA (5)-HF-Mg, while the PLLA droplets could completely cover the substrate without excessive accumulation on the surface of PLLA (10)-HF-Mg. The accumulation of droplets may make the surface uneven. Atomized drops are very small. Additionally, in this study, the thickness of the PLLA coating was controlled by controlling spraying times. Thus, on the surface of Mg alloy, PLLA drops may overlap to varying degrees. As spraying time increases, the heterogeneity will be significantly improved. Just as Figure 3 shows, the PLLA coating with 10 layers was more even than the PLLA coating with 5 layers. The thicker the coating, the more evenly the PLLA distributes, and the higher the elastic modulus and yield strength of the coating [45]. However, when the thickness of the coating is too thick, the adhesion of the polymer coating on the surface is worse [46]. A thicker coating may cause more inflammation because of more degradation products and longer degradation time. A thinner coating has smaller effects on the alloy’s mechanical property and faster endothelialization [47]. Considering the above, the PLLA coating was not sprayed with more layers in this study. Figure 5 shows that the thickness of the PLLA coating of PLLA (5)-HF-Mg and PLLA (10)-HF-Mg was 1.1 μm and 1.5 μm (Figure 4). This is in the range of thickness of currently used DES coatings. Moreover, the PLLA-coating thickness of PLLA (10)-HF-Mg was not twice as much as PLLA (5)-HF-Mg because multiple layers of PLLA provide partial re-dissolving instead of simple superposition. In this study, it can be seen that ultrasonic the atomization spraying method can easily make a coating as thin as chemical conversion and easily control the thickness by changing the PLLA layers.

The early-stage degradation behaviors of the coated and bare Mg alloy were observed using electrochemical tests and immersion tests. All of these three coatings reduced the degradation rate of Mg alloy according to the results of the Tafel test. I_corr_ can be used to evaluate the corrosion resistance of materials. The higher the I_corr_, the worse the corrosion performance. As Figure 7 and Table 2 show, the corrosion resistance of HF-Mg was significantly improved compared with the bare Mg alloy. Additionally, the corrosion resistance of PLLA (5)-HF-Mg and PLLA (10)-HF-Mg were also improved a little compared with HF-Mg. This may be related to the better corrosion resistance of PLLA itself and PLLA covering the small holes caused by hydrogen in the process of fluoride acid treating. This provided a physical barrier against water exposure. This result is consistent with the water contact angle in Figure 6 where PLLA-HF-Mg (5) and PLLA-HF-Mg (10) were more hydrophobic, which was due to the -CH_3_ hydrophobic group of PLA molecule chains [48]. Moreover, in this experiment, compared with PLLA (5)-HF-Mg, the corrosion resistance of PLLA (10)-HF-Mg was found not to be significantly improved, maybe because the electrochemical (or Tafel) method is more predictive for the initial corrosion rate of Mg-based materials than for long-term corrosion rate, such as 24 h or longer [49]. Anyway, all of the three coatings were more resistant to corrosion than the bare Mg alloy, as Figure 7 shows, which is similar with previous studies [50].

As a cardiovascular stent material, the test of long-term immersion is very important. SBF solution was used as the corrosion medium, which was similar to human plasma in inorganic ion concentration [51]. The surface of the bare Mg alloy was completely destroyed. The corrosion cracks found on the bare Mg alloy were deeper and more numerous than samples of coated Mg alloys. Of these three coatings, the cracks on the surface of PLLA (10)-HF-Mg were the slightest. Thus, it can be concluded that PLLA-HF-Mg (10) had the best property of corrosion resistance because the cracks were caused by the shrinking of the surface corrosion product during the drying procedure of the samples. Meanwhile, the degradation of Mg alloy led to another two changes: one is the release of ions and hydrogen to liquid and a change in pH; another is the change in element composition, which is shown in Figure 9 and Figure 10. Mg alloy can cause a rise in liquid pH [52]. The dense MgF_2_ coating reduced the contact between the Mg alloy and the liquid, thus reducing the degradation and lowering the pH. As Figure 14 shows, the PLLA/MgF_2_ coating could slow the degradation because when contacting the liquid, PLLA degraded first. While the degradation product of PLLA is acidic, the pH value of the medium for PLLA (5)-HF-Mg and PLLA (10)-HF-Mg was lower compared with HF-Mg. Since the immersion time was not long enough, the pH value of the medium for PLLA (5)-HF-Mg and PLLA (10)-HF-Mg did not have much difference. As for element content, after degradation, the content of all surfaces changed differently. As is known, the main corrosion products on Mg alloy’s surface are Mg(OH)_2_, MgO, MgCO_3_, CaCO_3_, and Ca_3_(PO4)_2_ [53]. The content of O can present the main corrosion product content, because O is the most abundant element in the corrosion products of Mg alloy. In this experiment, it could be seen that the content of O on the bare Mg surface was the most, which means severe corrosion. Comparing the content of F and Mg on the surface of all coated groups, it was found that on the surface of PLLA (10)-HF-Mg, there were nearly no F and Mg elements. Meanwhile, on the surface of PLLA (5)-HF-Mg, there were little F and Mg elements. This was perhaps due to the protection of the PLLA layer. Once immersed in the SBF, the PLLA protected the Mg alloy from contacting the liquid and degraded slowly.

### 4.2. The Novel Composite Coating Improves Biocompatibility of Mg Alloy

In this study, a dense, thin MgF_2_/PLLA coating was fabricated on Mg alloy to improve its property for the application of vascular stents. It was verified that the surface corrosion resistance of the Mg alloy was greatly improved by PLLA coatings and MgF_2_. However, it is not enough for vascular stents to have good mechanical properties and corrosion properties. It is also necessary to have good biocompatibility and be rapidly endothelialized [24]. In this experiment, HUVECs were selected as the research object to evaluate the cell adhesion and toxicity and the endothelium of modified Mg alloy. Two classical methods were used: one was the co-culture of HUVECs and Mg alloy; the other was the culture of cells in Mg alloy extract liquid [33]. In this experiment, it was found that HUVECs grew differently on different surfaces of Mg alloys. Compared with bare Mg alloy and HF-Mg, the PLLA coating provided a surface that was more conducive to HUVECs’ growth. HUVECs grew barely on the bare Mg alloy’s surface. Because of the absence of a protective layer, the bare Mg substrate underwent a quick corrosion process at the initial stage, leading to quick Mg^2+^ ion release, a sharp alkaline shift, as well as fast hydrogen liberation, which greatly influence cell growth [14]. The number of HUVECs on MgF_2_ was small, the same as the results of previous studies which concluded that endothelial cells on MgF_2_ were difficult to colonize [24]. However, in this experiment, there were a large number of cells on the PLLA/MgF_2_ coating, especially on the surface of PLLA (10)-HF-Mg. This may be related to the fact that PLLA could completely cover the surface, and the PLLA coating delayed the release of Mg^2+^, which is also consistent with the good biocompatibility of PLLA reported by many predecessors [25]. The cell viability of HUVECs cultured in coated Mg alloy extract liquid for 1 day was better than that of bare Mg alloy. While extracting for 3 days, only that of the PLLA (10)-HF-Mg group increased. This may because with the longer extracting time, the more degradation products accumulate, affecting the culture environment. In particular, the PLLA coating of PLLA (10)-HF-Mg completely covered the surface of Mg alloy, locked with the substrate tightly, and did not easily fall off.

When biological materials make contact with blood, hemolysis, coagulation, and even thrombosis may occur. Platelet adhesion is regarded as an essential reaction and will initiate thrombus formation and arterial occlusion [26]. Thus, it is essential to examine platelet adhesion. Platelets are divided into five different morphologies, reflecting different activation states of platelets: (I) round, (II) dendritic, (III) spread dendritic, (IV) spreading, and (V) fully spreading [54]. In this study, it was found that on the surface of the bare Mg alloy, the number of platelets was not only large, but also the morphology was deformed and spreading, which means the platelets were activated. Conversely, on the surface of HF-Mg, PLLA (5)-HF-Mg, and PLLA (10)-HF-Mg, the amounts of platelets decreased and became slightly pseudopodia spreading. In particular, rare platelets adhered to the surface of PLLA (10)-HF-Mg and the platelets were nearly round. The results suggested that platelets which adhered to the PLLA (10)-HF-Mg surfaces were less and less activated than other samples. Similar observations have been shown in previous studies that pure Mg had more adherent platelets compared with other Mg–1X alloys [55]. Researchers suspected that hydrogen, released during the platelet adhesion test, may promote platelet adhesion because increased gas nuclei may affect the proteins and direct contact with platelets [56]. Meanwhile, rapid corrosion caused an increase in the surface area, leading to increased plasma protein adsorption, and subsequent platelet adhesion and activation [15].

## 5. Conclusions

In order to solve the problem of rapid degradation and insufficient biocompatibility, this study creatively proposed the fabrication of a novel composite coating. A MgF_2_ layer was fabricated on Mg alloy, then a PLLA coating was sprayed on HF-Mg via ultrasonic atomization spraying. Moreover, in this experiment, different layers of PLLA were compared. This novel composite coating was studied from the perspective of its characteristics, corrosion resistance, endothelialization, cell compatibility, and hemocompatibility. The mechanism of controlling the degradation rate and enhancing biocompatibility was discussed in this study. The conclusions are drawn as follows:The MgF_2_ layer was smooth and compact, but had some holes. The PLLA coating could partly cover these holes, providing a good physical barrier and reducing the corrosion of the Mg alloy substrate. The PLLA coating with five layers was uneven and could not cover the surface. However, the PLLA coating with 10 layers could almost cover the surface without excessive droplet overlap, mechanically locked with the substrate.Compared with the I_corr_ of the bare Mg alloy (9.58 × 10^−4^ A/mm^2^), the I_corr_ of PLLA (10)-HF-Mg (7.0 × 10^−5^ A/mm^2^) was significantly decreased, indicating a great improvement in corrosion resistance. While all the samples were immersed in SBF for 14 days, PLLA (10)-HF-Mg also had the best corrosion resistance. There were nearly no cracks on PLLA (10)-HF-Mg and some corrosion cracks on PLLA (5)-HF-Mg. However, the cracks on the bare Mg alloy were numerous and deep.HUVECs barely grew on the bare Mg alloy surface, but grew very well on the PLLA (10)-HF-Mg surface, indicating the good biocompatibility and endothelialization of PLLA (10)-HF-Mg. Meanwhile, the RGR of HUVECs in the PLLA (10)-HF-Mg extract significantly increased by 73% (1 day) and 111% (3 days).With the new composite PLLA/MgF_2_ coating, the number of adherent platelets was decreased. The adherent platelets showed a nearly round shape, which indicates less activation.

In conclusion, the PLLA/MgF_2_ coating, especially the coating of PLLA (10)-HF-Mg, successfully controlled the Mg alloy degradation, resulting in better biocompatibility, including better endothelization and hemocompatibility. Additionally, it was easy to control the degradation rate of Mg alloy by controlling different layers of PLLA. Therefore, designing composite PLLA/MgF_2_ coatings using the ultrasonic atomization spraying method makes it a promising candidate for the production of cardiovascular stents and deserves further study.

## Figures and Tables

**Figure 1 materials-16-00682-f001:**
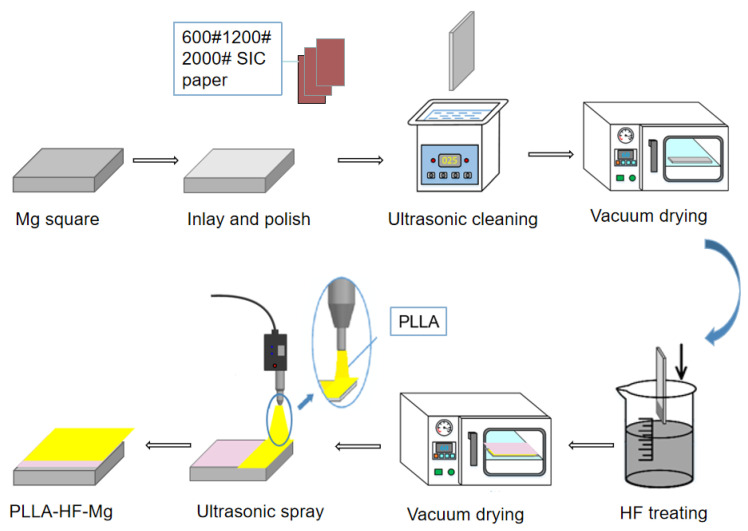
Preparing process of samples.

**Figure 2 materials-16-00682-f002:**
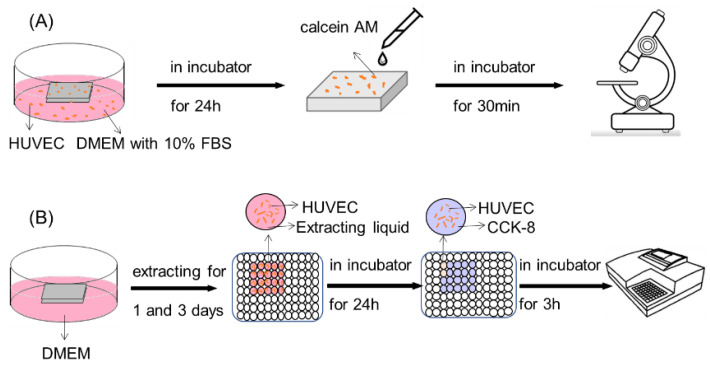
Cell testing of coated and bare Mg alloys (**A**) Processing of testing cell adhesion, (**B**) Processing of testing cell viability.

**Figure 3 materials-16-00682-f003:**
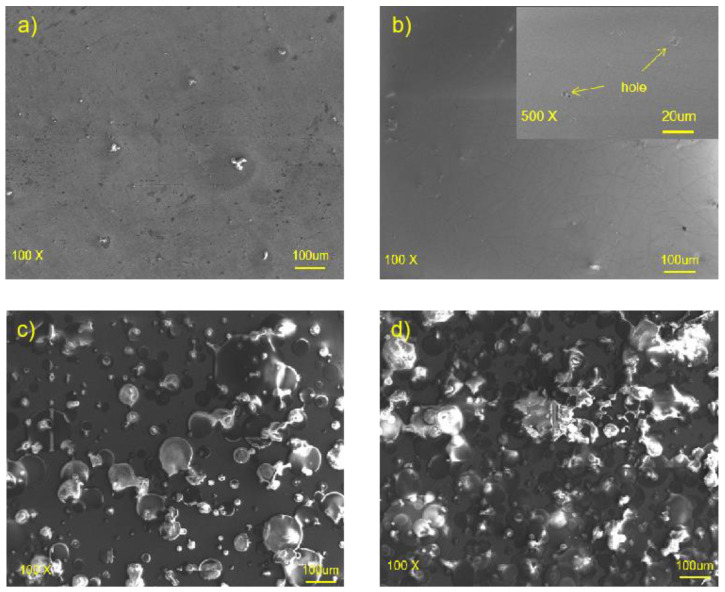
SEM images of (**a**) bare Mg alloy, (**b**) HF-Mg and (**c**) PLLA (5)-HF-Mg, (**d**) PLLA (10)-HF-Mg.

**Figure 4 materials-16-00682-f004:**
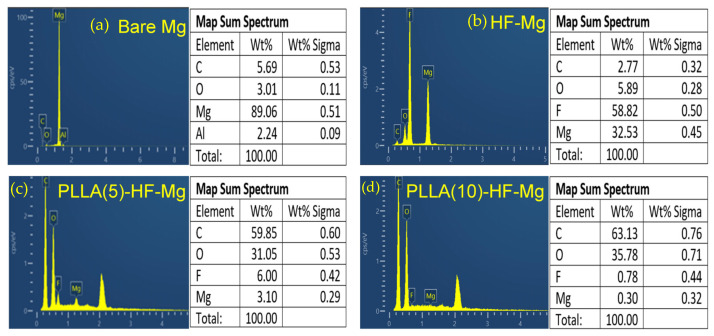
Elemental composition of the surface of (**a**) bare Mg alloy, (**b**) HF-Mg and (**c**) PLLA (5)-HF-Mg, (**d**) PLLA (10)-HF-Mg.

**Figure 5 materials-16-00682-f005:**
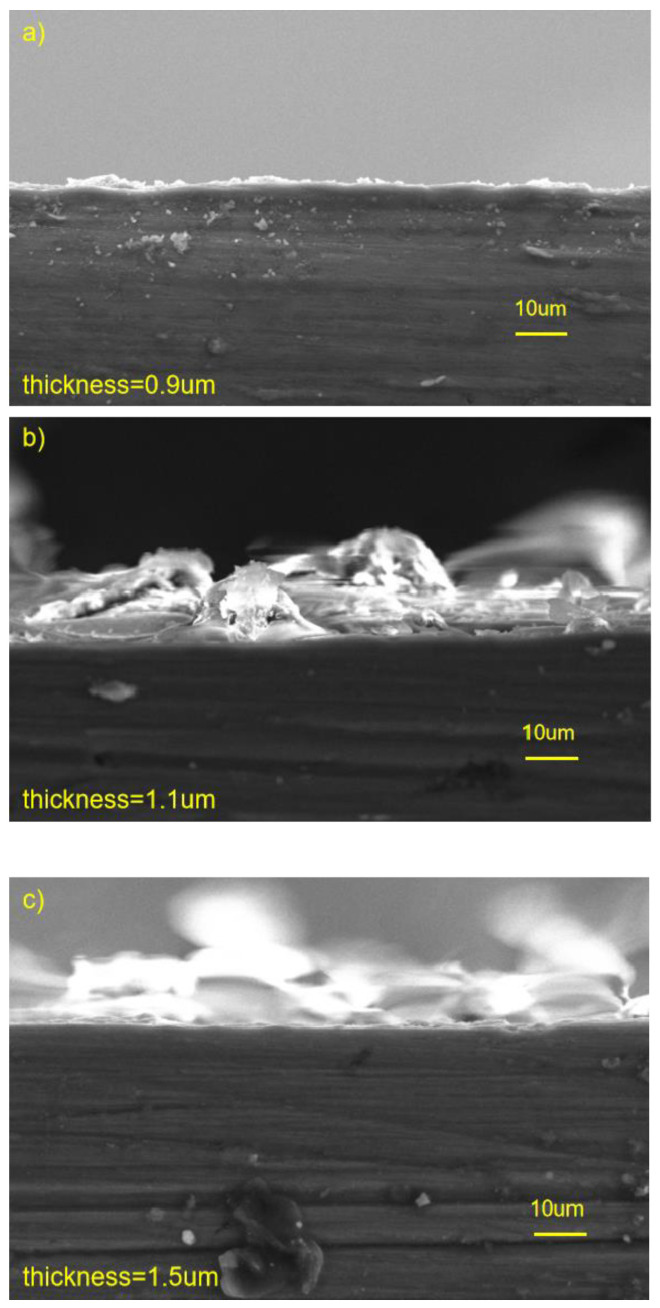
SEM micrograph of the cross-section of three coated samples (**a**) HF- Mg, (**b**) PLLA (5)-HF-Mg, (**c**) PLLA (10)-HF-Mg.

**Figure 6 materials-16-00682-f006:**
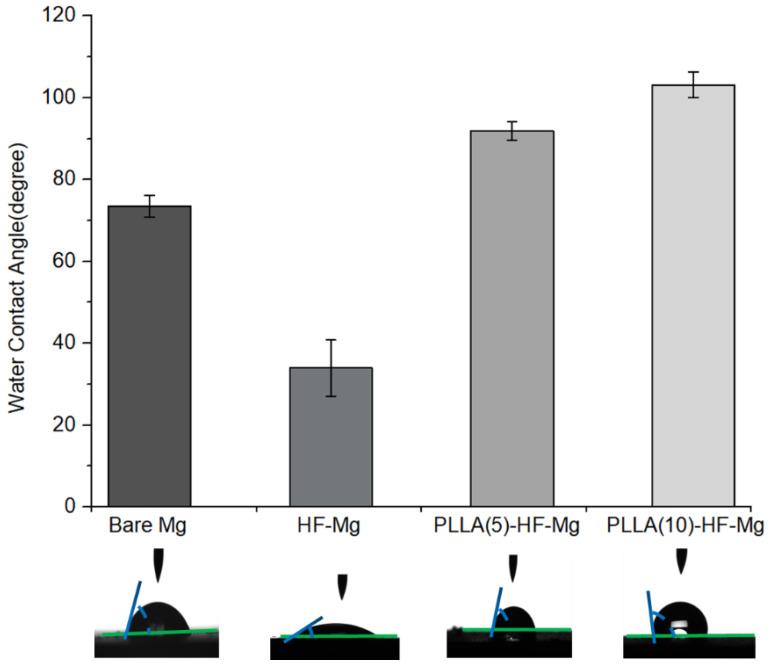
Water contact angle of coated and bare Mg alloys. Values are mean ± standard deviation (*n* = 3).

**Figure 7 materials-16-00682-f007:**
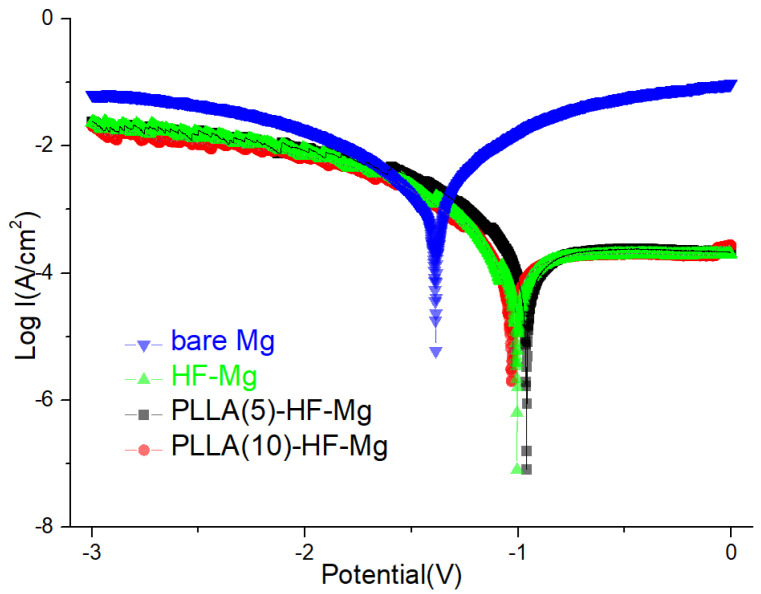
PDP curves of coated and bare Mg alloy samples in SBF.

**Figure 8 materials-16-00682-f008:**
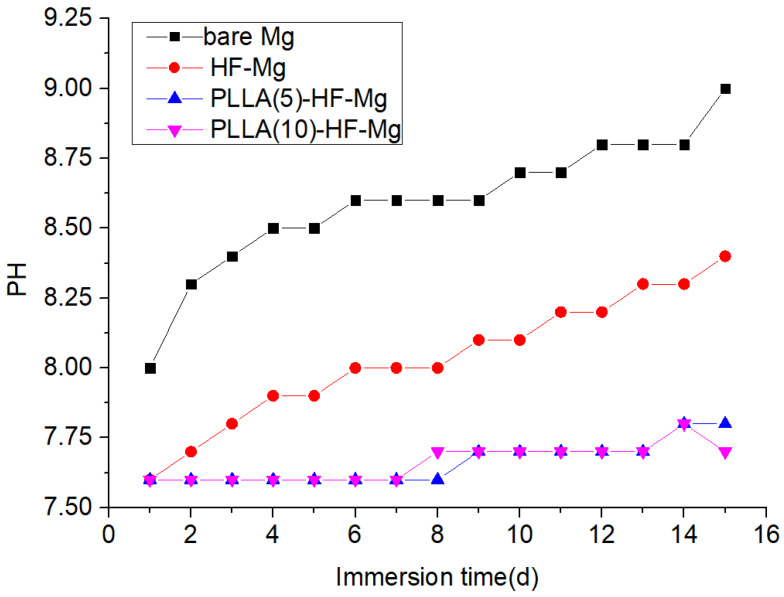
pH values of SBF solution for coated and bare Mg alloy over time after immersion.

**Figure 9 materials-16-00682-f009:**
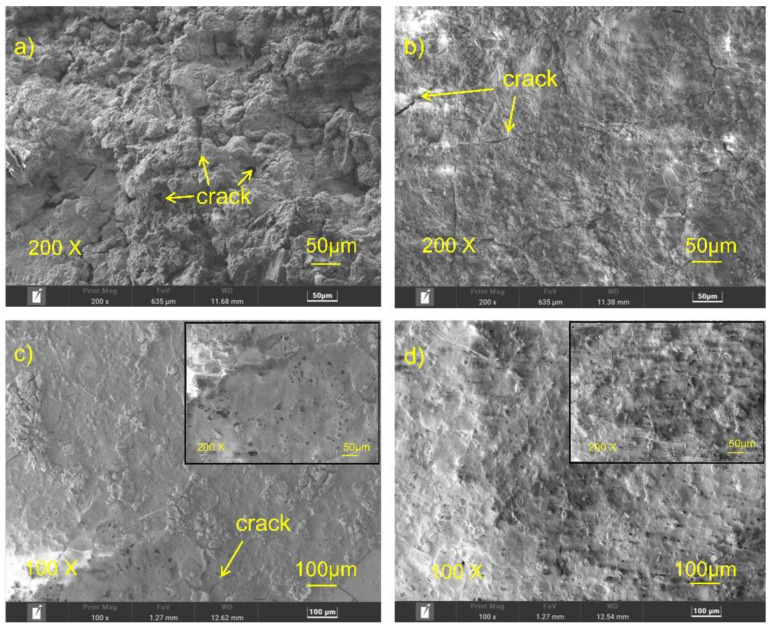
SEM image of all samples after immersion in SBF for 15 days (**a**) bare Mg, (**b**) HF-Mg and (**c**) PLLA (5)-HF-Mg, (**d**) PLLA (10)-HF-Mg.

**Figure 10 materials-16-00682-f010:**
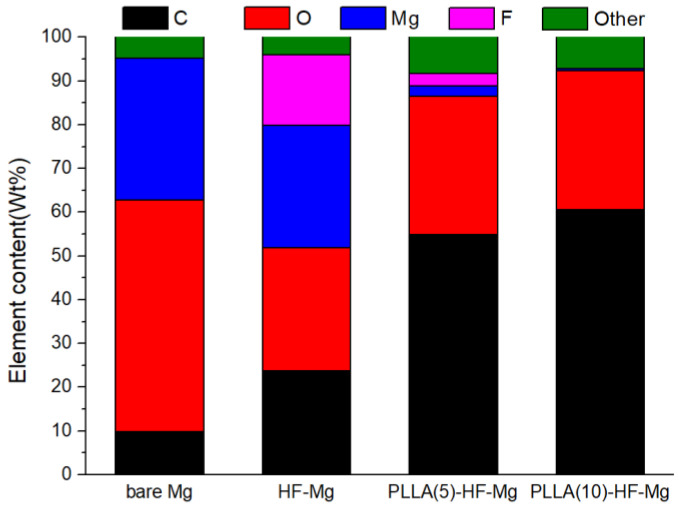
Corrosion surface components of all samples after immersion in SBF for 15 days analyzed by EDS.

**Figure 11 materials-16-00682-f011:**
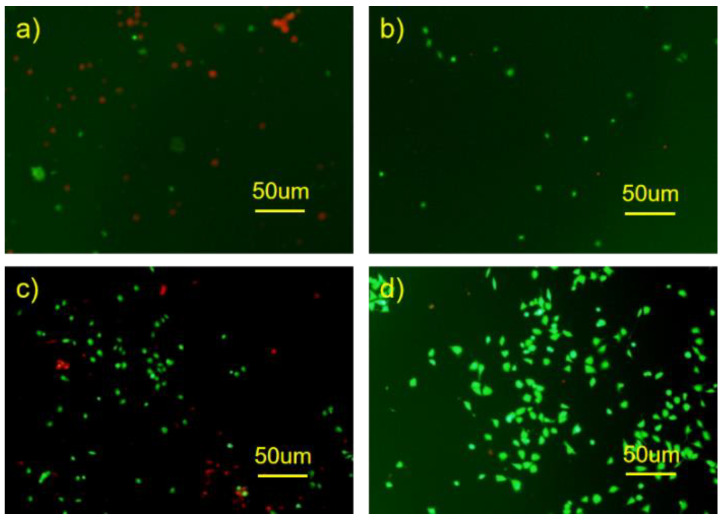
Fluorescence images of HUVECs attached directly on bare Mg and coated Mg alloy after in vitro culture for 24 h, where live cells fluoresce green and dead cells fluoresce red ((**a**) bare Mg (**b**) HF-Mg (**c**) PLLA (5)-HF-Mg (**d**) PLLA (10)-HF-Mg).

**Figure 12 materials-16-00682-f012:**
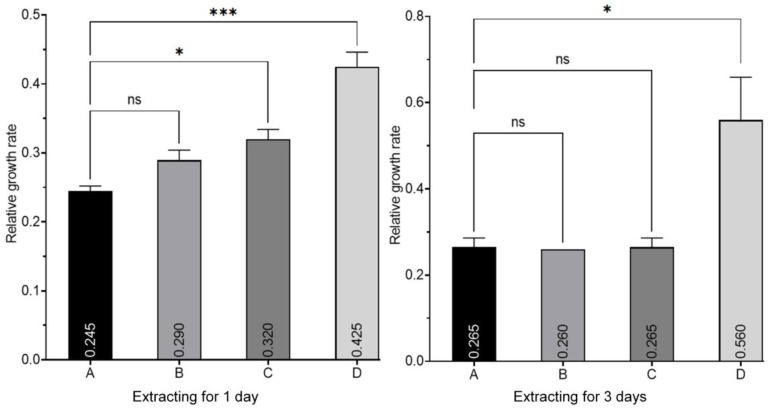
In vitro cell vitality tests of Mg alloy extracts on HUVECs. (A: bare Mg alloy; B: HF-Mg; C: PLLA (5)-HF-Mg; D: PLLA (10)-HF-Mg). Values are mean ± standard deviation (*n* = 3). * *p* < 0.05, *** *p* < 0.001. ns: no significance.

**Figure 13 materials-16-00682-f013:**
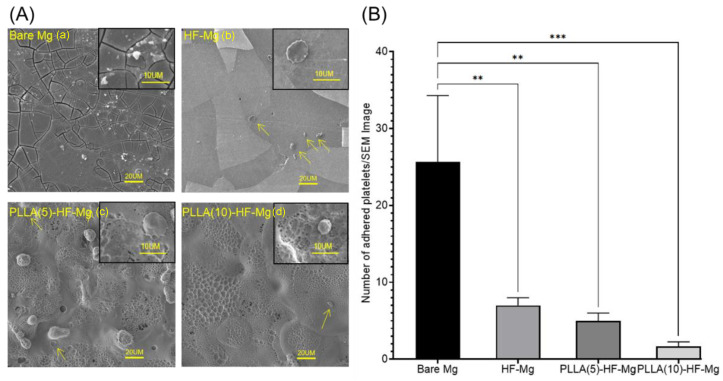
(**A**) Platelet adhesion tests of (**a**) bare Mg alloy (**b**) HF-Mg (**c**) PLLA (5)-HF-Mg (**d**) PLLA (10)-HF-Mg. (**B**) Number of adhered platelets/SEM Image, (** *p* < 0.01, *** *p* < 0.001).

**Figure 14 materials-16-00682-f014:**
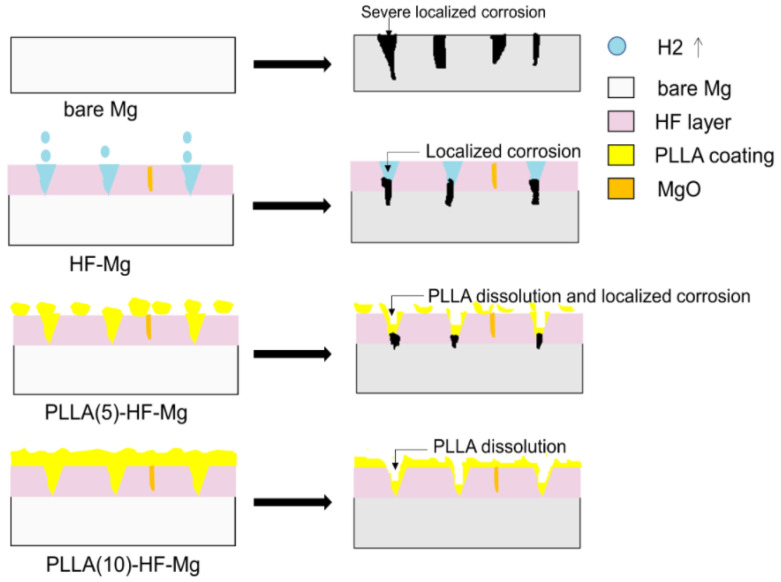
Mechanism of Mg alloys’ degradation controlled by different coatings.

**Table 1 materials-16-00682-t001:** SBF preparation.

Sequence	Name	Quantity
1	NaCl	8.035 g
2	NaHCO_3_	0.355 g
3	KCl	0.225 g
4	K_2_HPO_4_·3H_2_O	0.231 g
5	MgCl_2_·6H_2_O	0.311 g
6	1.0 M-HCl	39 mL
7	CaCl_2_	0.292 g
8	Na_2_SO_4_	0.072 g
9	Tris	6.118 g
10	1.0 M-HCl	0–5 mL

**Table 2 materials-16-00682-t002:** Corrosion current density of coated and bare Mg alloy samples (unit: μA/cm^2^).

Samples	Bare Mg	HF-Mg	PLLA (5)-HF-Mg	PLLA (10)-HF-Mg
I_corr_	958 ± 5	143 ± 3.5	81 ± 1.8	70 ± 3.5

Note: All experiments were conducted in triplicate.

## Data Availability

The data used to support the findings of this study are available from the corresponding author upon request.

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
