# Peer review of "A Novel PLLA/MgF2 Coating on Mg Alloy by Ultrasonic Atomization Spraying for Controlling Degradation and Improving Biocompatibility"

_materials, 2023, doi:10.3390/ma16020682_

Round 1

Reviewer 1 Report

In this article, Peng et al have fabricated and characterized Mg-F2 alloy coated with 5 or 10 layers of PLL. This an excellent research article. The authors should address the followings:

1.       Numerous sentences are grammatically incorrect and must be corrected.

2.       There is sudden appearance Error messages at several places in the text.

3.       What is the composition and source of SBF?

4.       Figure 3; there is extensive heterogeneity of the PLL deposition. Please comment on it.

5.       What stent materials are currently used, and how the Mg-F-PLL compare with them.

6.       Platelet activation should be checked with molecular markers.

7.       Aggregation should be determined by some molecular markers.

Reviewer 2 Report

This version of the manuscript does not look worthy and cannot be recommended for publication in this form and at least needs a major revision.

1.     Considering MgF2. it must be briefly mentioned that its functional properties and their degradation are highly dependent on point structural defects. Even in crystals, point defects play a significant, and in many cases simply decisive. See, for example:

Lisitsyn, V. M., et al (2016). Stabilization of primary mobile radiation defects in MgF2 crystals. Nuclear Instruments and Methods in Physics Research Section B: Beam Interactions with Materials and Atoms374, 24-28.

https://doi.org/10.1016/j.nimb.2015.08.002

2.     Paragraph 2.1 and 2.2. In the manuscript , there are several messages, like  s Error! Reference source not found.”  This does not allow you to check the accuracy of the information due to the unavailability of the corresponding link.

3.     Fig. 6 and Table 1 need error bars and corresponding notes in the text.

4.     The quality of Fig.9 is not enough.

5.     Henceforth, authors should carefully check the text before they upload it. The text and the list of references are not connected in any way. Therefore, the adequacy of the citation cannot be established.

Reviewer 3 Report

1. Authors must read the journal style guide; the numbers of references must be 1, 2, 3 ... etc.
2. Page 2: Please add the ATP abbreviation description.
3. The article is difficult to read because of broken references ("Error! Reference source not found"). It is not good practice. The authors treated their manuscript very carelessly, which is unacceptable. It was enough just to generate a PDF file and view it before submission.
4. Section 2.1: The Mg substrate preparation procedure is unclear. Why are the samples cleaned, polished, and again cleaned?
5. Page 5, section 2.5: 100 ul = 100 μL? Please check.
6. Please check and use the subscript symbols where it is appropriate. For example for corr in Icorr.
7. Fig. 7, Table 1: The corrosion current designated as I, J or E?
8. Section 4 is missed.
9. Page 14: What does "... the coating of the elastic modulus and yield strength."?

Round 2

Reviewer 2 Report

After quite a successful revision, this manuscript can be recommended for publication.